# Strategies to Cope with Inferior Long-Term Photostability of Bentonite Polyolefin Nanocomposites

**DOI:** 10.3390/polym16040535

**Published:** 2024-02-17

**Authors:** Erik Westphal, Guru Geertz, Michael Großhauser, Elke Metzsch-Zilligen, Rudolf Pfaendner

**Affiliations:** Division Plastics, Fraunhofer Institute for Structural Durability and System Reliability, 64289 Darmstadt, Germanyguru.geertz@lbf.fraunhofer.de (G.G.); michael.grosshauser@lbf.fraunhofer.de (M.G.); elke.metzsch-zilligen@lbf.fraunhofer.de (E.M.-Z.)

**Keywords:** polyethylene, EVA, organo-modified layered silicate, bentonite, nanocomposite, ATH, long-term photostability, weathering, durability

## Abstract

This study provides insight into the causes of inferior long-term stability of nanocomposites based on organic layered silicates (OLSs) used for cable mantles. A hierarchy was established by analyzing bentonite products and their respective polyolefin nanocomposites. Thermogravimetric analysis (TGA), X-ray diffraction (XRD), gas adsorption, energy-dispersive spectroscopy (SEM-EDX), and infrared spectroscopy (IR) provided evidence for the adsorption of stabilizers onto the filler surface and thus their reduction in activity, promoting polymer oxidation. This behavior corresponds to the specific surface area of the incorporated OLS. Therefore, it can be stated that gas adsorption and XRD are especially useful for the evaluation of long-term photostability. It was revealed that photocatalytically active iron is of secondary importance since iron-rich bentonites still formed the most stable nanocomposite. This also applies to the Hofmann elimination products of the modifying agent, where higher contents do not accelerate the degradation process. No elimination products could be traced within the composites. Due to the polymer-filler interface being essential for long-term photostability, prior analysis of the filler surface properties can be used to estimate the stability of the respective nanocomposite as a rationale for product selection in the early stages of development. The reasons identified in this work for decreasing the long-term photostability of OLS nanocomposites compared with unfilled formulations is an important step toward increasing their stability.

## 1. Introduction

The incorporation of organically modified layered silicates (OLSs) with a high aspect ratio into polymers to form nanocomposites comes with a wide range of benefits, such as improved mechanical properties, like tensile strength and elongation, barrier properties, optical properties, and flame retardancy [1,2,3,4,5]. This improvement is mostly attributed to the vastly increased specific surface area of the filler due to its dispersion on a nanometer scale [5]. However, the distribution of the nanofiller in polyolefins also leads to a significantly reduced long-term thermal and photostability. This cripples the range of applications for polyolefin–OLS nanocomposites despite their otherwise desirable properties [6,7,8,9,10,11,12,13,14].

Common hypotheses for the decreased stability are degradation reactions of the organo-modifier in the melt processing steps [5,15], the photocatalytic activity of transition-metal ions and other species incorporated in the silicate [16], or the adsorption and, thus, inhibition of stabilizers onto the surface of the OLS [17,18]. Whilst each of these factors has been found to contribute to compound destabilization to some extent, a comprehensive answer concerning the relative importance of these factors has not yet been established. Research has been undertaken to improve the long-term thermal and photostability of OLS nanocomposites utilizing specialized additives [19,20,21] or direct functionalization of the OLS [4,22]. However, little has been undertaken to understand the impact of the inherent material properties of the OLS on long-term composite photostability. A study on nano-dispersed pigments concluded that a significant impact is generated by the size and surface area of the pigment [23], implying that similar results can be found for OLS nanocomposites. The recent adaptation of a harmonized norm EN 50575:2017 concerning the flammability of cables prompted further investigation of this topic, specifically for polyolefin-based materials with ATH as a flame retardant and OLS as a synergist.

The filler–polymer interface in OLS nanocomposites is not easily accessible with most analytical methods. Therefore, approximations of the interactions via model experiments might be the only way to explain the behavior of OLS nanocomposites and are, therefore, explored in this work.

The aim of this research was twofold: firstly, to study the characteristics of OLS nanocomposites to elucidate the mechanisms that lower their long-term photostability, and secondly, to lay the groundwork for a rational upfront selection of OLSs to contribute to the early stages of the development of OLS nanocomposites with enhanced long-term photostability. Four different, commercially available bentonite products were tested for their potential use in a model formulation for cable applications: three different types of OLS bentonite, each modified with the same organic salt, dimethyl-dihydrogenated tallow ammonium chloride (DMDHT), as well as natural bentonite without a modifier as reference.

## 2. Materials and Methods

The bentonite grades used in this study were supplied by BYK-Chemie GmbH, Wesel, Germany (Cloisite 116, Cloisite 20A, BYK MAX CT 4260) and Laviosa, Livorno, Italy (Dellite CW9). Except for the unmodified Cloisite 116, the bentonites were organically modified by DMDHT via cation exchange. Nitrogen gas adsorption analysis was performed at 77 K with a Thermo Fisher Scientific Surfer Brunauer–Emmett–Teller analyzer. The samples were dried for two days at 353 K in vacuo (<10^−5^ bar). X-ray diffraction measurements (XRD) were performed using a D4 Endeavor by Bruker, at 40 kV, using CuKα radiation (λ = 0.179 nm) at a scan rate of 2°·min^−1^. Pure bentonite was analyzed as a powder, and the composites as plates. SEM imaging of gold-coated bentonites on an adhesive carbon film was carried out with a Supra 25 by Zeiss at a magnification of 500× at 8 kV. The presence or absence of silicate tactoids or exfoliated layers was investigated by transmission electron microscopy (TEM) with an EM10 TEM by Carl Zeiss AG at a magnification of 80,000× with 40 nm thin sample slices cut with a diamond knife at −60 °C using a 1:1 solution of DMSO and water. Energy-dispersive spectroscopy (SEM-EDX) was applied to obtain the elemental composition of bentonite products, focusing on iron. For this, a Philips XL30 FEG HREM was used in conjunction with an EDAX Genesis EDX system on compacted bentonite platelets. To determine the respective content of the organo-modifier via thermogravimetric analysis (TGA), a Mettler Toledo TGA2 was utilized. The samples were subjected to a temperature ramp (110 to 800 °C, with a gradient of 10 K·min^−1^) in aluminum oxide crucibles in an atmosphere of synthetic air.

For compounding, EVA (Escorene^TM^ Ultra UL 00328; 22 parts) was supplied by ExxonMobil, Irving, Texas, and LLDPE (LL318; 10 parts) by Braskem, Sao Paulo, Brazil. Two compatibilizers, a terpolymer by Arkema, Colombes, France (Lotader^®^ 3210; 4 parts), and PE-g-MA by DuPont, Wilmington, U.S.A. (Fusabond^®^ E226; 4 parts), were used. For stabilization, a hindered phenol antioxidant (ADK Stab AO-60; 0.2 parts; Adeka, Arakawa, Japan) was used and as light stabilizers, BASF’s (Ludwigshafen, Germany) Chimassorb^®^ 944 and 81 (0.1 parts each) were used alongside the flame retardant ATH supplied by Nabaltec AG, Schwandorf, Germany (Apyral^®^ 40CD; 62 parts). Lastly, 4 parts of bentonite were added.

Compounding was performed using a co-rotating twin screw extruder Leistritz MIC 27 at 165 °C and 300 rpm. Three gravimetric feeders were used, two of which were fed into the main extruder feed: the polymer–additive mix and half of the ATH. The side feed was fed with the second half of the ATH and the bentonite. The air-cooled polymer strand was then granulated for use in an Arburg Allrounder 320 C Golden Edition injection molding machine with a tool temperature of 40 °C and a mass temperature of 190 °C to produce dumbbell-shaped samples of type 5A according to DIN EN ISO 527-2.

Weathering was accomplished using a QUV/spray weathering chamber by QLAB equipped with UVA-340 lamps. A cycle of 10 h irradiation with 0.68 W·m^−2^ at 340 nm, 50% rel. humidity, a 60 °C black panel temperature, and 2 h of darkness with condensing humidity at 50 °C was applied for a total of 3000 h. Samples were collected in intervals of 500 h.

Flame retardancy was evaluated at 35 kW·m^−2^ with a Cone Calorimeter Pro I by Wazau conforming to ISO 5660-1. The sample plates of 100 × 100 × 3.2 mm^3^ were injection-molded under identical conditions to the type 5A samples and conditioned for 7 days at 23 °C and 50% relative humidity prior to measurement.

The composite stability was investigated using ATR-FT-IR-spectroscopy, with a Nicolet Nexus 670 by Thermo Fisher Scientific equipped with a Golden Gate-ATR inset. Thirty-two accumulations of scans were collected between 4000 and 650 cm^−1^ at a 4 cm^−1^ scanning resolution with a separate background correction before every spectrum. The averaged carbonyl index (CI) was used by measuring two distinct sample regions (at the full width of the dumbbell shape and at the taper) and employing the following Equation (1):(1)$$CI=A_{3040-2740}/A_{1850-1600},$$
where *A* represents the area enclosed between two wavenumbers.

## 3. Results and Discussion

### 3.1. Thermal Analysis of Bentonites

The thermal decomposition behavior of the modified bentonites as determined by TGA showed additional mass loss steps compared with the unmodified ones (Figure 1). The lowest onset for DMDHT degradation extracted from the DTG curves was observed in Dellite CW9 at 203 °C, whereas the onset for MAX CT 4260 and Cloisite 20A lay at 210 °C. Dellite CW9 also exhibited a high amount of degradation early on, whereas this point was reached at higher temperatures for the other OLS. The two maxima in the DTG curves for Dellite CW9 indicate the partial enrichment of DMDHT on the surface of Dellite CW9 particles, not partaking in the interlayer penetration of the silicate via cation exchange. Similarly, Bertin et al. observed differing degradation onsets for Irganox 1010 on talc, depending on whether the stabilizer was free, bound to the surface, or fully adsorbed [24]. Hence, the DMDHT on the surface was decomposed first with a maximum at 288 °C, and the organic modifier embedded into the interlayers decomposed with a maximum at 333 °C. No such clear separation was observed for the remaining two OLSs, where only one degradation maximum was visible at 311 °C. Dellite CW9 showed by far the highest amount of organic modifier, followed by the other two OLSs with comparable loadings of the modifier. The next step in thermal degradation occurring between 450 and 650 °C was observed for all the OLSs as well as the unmodified Cloisite 116 and may, therefore, be linked to an inorganic decomposition mechanism. This step can, indeed, be attributed to the dehydroxylation of silicate-bound structural OH units in montmorillonite, the main constituent of bentonite [25]. The remaining step for mass loss at temperatures higher than 650 °C was observed for OLS only and most likely corresponded to some charring process in the organic residues.

### 3.2. Adsorption Analysis of Bentonites

The results of the BET analysis of the nitrogen adsorption measurements are compiled in Table 1. The unmodified Cloisite 116 exhibited the highest specific surface area with 29.788 m^2^·g^−1^, almost twice the area of all the OLSs. The process of cationic exchange seemed to limit the surface area by filling the pores that would otherwise be accessible for gas adsorption with the organic modifier, decreasing the pore volume compared with the unmodified silicate. The TG results seem to be in line with the results of gas adsorption, proving that the highest content of organo-modifier also leads to the highest loss of pore volume. If the adsorption of additives onto the bentonite surface plays an important role in lowering the long-term photostability of polymer composites, both surface area and pore diameter should be important properties to probe the magnitude of such adsorption phenomena. A lower total surface area statistically allows for fewer adsorption events, whilst a lower diameter hinders the often sterically demanding stabilizer molecules from interacting with the respective pores. Considering this theory, Dellite CW9 could be a candidate for the optimal choice of a polymer nanocomposite with improved long-term photostability, exhibiting one of the lowest measured surface areas at 15.40 m^2^·g^−1^ and the lowest pore diameter at 11.65 nm on average among all OLSs as well as the lowest pore volume at 0.0363 cm^3^·g^−1^, almost half that of the other OLSs. The organic modifier that penetrates the interlayers of the silicate obviously increases the pore diameter of the resulting OLS, indicating that the pores accessible for adsorption are directly related to the interlayer space. Additional organic modifier, like in the case of Dellite CW9, can close those pores, thus lowering the surface area and the pore volume.

### 3.3. Particle Analysis of Bentonites

To exhibit the barrier properties that are widely accepted to be the main contributor to the flame retardant effect of layered silicate composites [26], sufficient dispersion of the bentonite platelets is necessary [5,27,28], and the use of a polymeric compatibilizer as well as an organic modifier in the silicate interlayers is necessary to overcome the entropic barrier [6]. However, there seems to be a limit to improving the flame retardancy, regarding either the amount of added OLS or the degree of dispersion [27]. Therefore, there should be an optimal degree of dispersion, where the barrier properties on the burning material are fully established. Levels of dispersion below that are not able to enhance the flame retardancy to a satisfactory level, whereas an OLS with exceedingly high degrees of dispersion results in a higher specific surface area, giving more room to interaction with stabilizers, possibly deactivating them in the process. The particle size of bentonite products may be beneficial or detrimental to this process. Smaller particles can be well dispersed in the composite with a lower amount of shear forces yet also tend to exfoliate easily, resulting in a high specific surface area. SEM imaging of bentonite products (Figure 2) was used to determine the average particle size of each product.

Using ImageSP Viewer, the diameters of every particle were measured vertically since they did not have a uniform shape that would dictate a default measurement direction. No preferred alignment was induced by the sample preparation, allowing for any consistent direction to be representative. The respective histograms of the particle size distributions are displayed in Figure 3, and additional data are compiled in Table 2. Cloisite 20A had the smallest particles overall, as well as a decently narrow distribution. The maximum of the particle size distribution at 10.02 µm was, indeed, well below every other examined bentonite, followed by Dellite CW9 at 12.38 µm, the unmodified Cloisite 116 at 14.45 µm, and, lastly, the largest average particle size in MAX CT 4260 at 15.88 µm. The full width at half maximum of the particle size distributions confirmed the narrowest distribution for Cloisite 20A, whereas MAX CT 4260 had the widest distribution. Judging from these data, Cloisite 20A should give rise to the best dispersion in a composite, given that the compounding process can achieve such a level of dispersion. The high amount of organic modifier inside the galleries of Cloisite 20A facilitates the formation of the largest interface and should potentially destabilize the composite the most.

### 3.4. Interlayer Distances of Bentonites

Figure 4 depicts the XRD signals of the pure bentonite samples. The unmodified Cloisite 116 exhibited its *d_001_* peak at 7.1°, corresponding to an interlayer distance of 12.4 Å. [27,29,30]. This distance was increased for the OLS during cation exchange by inserting the sterically demanding DMDHT molecules into the gallery, resulting in an average interlayer distance of 24.9 Å for MAX CT 4260 and a slightly larger distance for Cloisite 20A of 25.6 Å, all being consistent with similar OLSs. [31] This indicates that the latter tends to intercalate more easily since the penetration of the interlayer by polymer chains is facilitated by the larger interlayer, the higher amount of organic modifier acting as a compatibilizer and the smaller particle size to begin with. In comparison, Dellite CW9 displayed a unique distribution of interlayer distances with two base-separated peaks. One peak corresponded to an increased *d_001_* distance of 27.6 Å, the highest among the examined OLSs, facilitating intercalation and the creation of a barrier for enhanced flame retardancy. In addition, Dellite CW9 showed a second portion of bentonite with a smaller interlayer distance of 18.5 Å. This portion with a smaller interlayer distance, whilst still being well-compatible with the polymer due to the organic modifier, is bound to create less intercalated structures in the resulting composite. A small fraction of unmodified bentonite remained for all OLSs, giving rise to a refraction at 7.1°. The combination of different interlayer distances in Dellite CW9 may be a good prerequisite to form an effective flame retardancy barrier whilst also minimizing the specific surface area of the filler to lower interactions with the stabilizers.

### 3.5. Long-Term Photostability of Flame-Retardant Bentonite Nanocomposites

To evaluate the hypothesis established by TG analysis, gas adsorption, SEM imaging, and XRD, compounds containing the respective bentonite products as well as a compound without any silicate filler were produced. The XRD diagrams are shown in Figure 5. The low amount of bentonite in the composites combined with the high amount of ATH present impeded the interpretation of the resulting signals due to low signal intensity. Whilst the Cloisite 116 composite produced no noticeable peaks in the XRD, the presence of silicate tactoids was confirmed via TEM micrographs, ruling out the possibility of the full exfoliation of the silicate. The Cloisite 20A composite depicted a peak corresponding to the highest interlayer distance for all OLS composites of 38.8 Å, combined with a sharp rise in the intensity bordering the 2θ angle of 1°, implying the formation of strongly intercalated structures and possibly exfoliated structures with a high degree of separation between silicate layers. [27] The Dellite CW9 composite depicted a peak corresponding to a slightly smaller interlayer distance of 37.58 Å and was not followed by an increased signal intensity bordering 1°, indicating the degree of intercalation to be lower compared with Cloisite 20A. Lastly, the MAX CT 4260 composite produced a linear rise in the signal intensity between 3 and 1°, whereas the other composites showed a valley. Combined with the findings of the tactoids as well as exfoliated silicate layers in the TEM micrographs for all three OLS composites, an increase in the interlayer distance with a wider spread compared with Dellite CW9 and Cloisite 20A can be concluded for the MAX CT 4260 composite.

Cloisite 20A possessed the highest observed specific surface area with a high number of high-volume pores as well as an overall smaller particle size, which are factors enabling a good dispersion of the silicate in the polymer matrix since they facilitate an increase in the interlayer distance, synonymous with the penetration of the interlayer by surrounding polymer chains. The result is an increased specific surface area of the silicate filler in the composite, potentially enabling interactions like the adsorption of the stabilizer that decreases the long-term stability to take place more prominently. Hence, the usage of Dellite CW9 should facilitate obtaining the highest long-term photostability due to its low specific surface area in pristine form, including small pores of a low volume, as well as its composited interlayer distance leading to a lower degree of intercalation compared with Cloisite 20A.

To examine the impact of the respective bentonite samples on the flame retardancy of the respective compounds, cone calorimetry was applied. Significant values like the peak heat release rate (PHRR), the total heat and smoke release (THR/TSR), as well as the maximum average rate of heat emission (MARHE) were lowered by the addition of any bentonite, whilst the rate of mass loss was decreased, resulting in a greater amount of residue (Table 3). The PHRR for the silicate-free composite amounted to 182 kW·m^−2^ and was almost halved by the addition of any OLS. Whilst Cloisite 20A, Dellite CW9, and MAX CT 4260 decreased the PHRR to 100, 92, and 102 kW·m^−2^, respectively, the addition of unmodified Cloisite 116 resulted in a smaller decrease to 113 kW·m^−2^. The heat released during combustion was most successfully decreased by the silicate barriers produced by the organo-modified bentonites. The values for THR, TSR, and MARHE depict a difference between MAX CT 4260 and the other OLS. All of them improved the flame-retardant performance of their composite compared with the silicate-free composite and even the Cloisite 116 composite. Peculiarly, MAX CT 4260 always performed worse than Cloisite 20A or Dellite CW9 and with a significantly higher standard deviation, stemming from a less consistent flame retardancy. This was likely caused by an uneven distribution of MAX CT 4260 in the composite, resulting in visible agglomerates of the filler on the unburnt sample surface.

The MARHE is a parameter used to approximate flammability in a single figure and reflects the results of other parameters obtained by cone calorimetry. [32] Whilst the usage of unmodified bentonite could not lower the MARHE to the same extent as the OLS, it improved the MARHE from 95.5 ± 14.7 to 74.3 ± 1.4 kW·m^−2^. For the addition of MAX CT 4260, the MARHE was lowered to 59.4 ± 25 kW·m^−2^. A better flame retardancy was obtained by adding Cloisite 20A (57.5 ± 2.4 kW·m^−2^) or Dellite CW9 (52.9 ± 6.1 kW·m^−2^) with very similar results. Overall, Dellite CW9 and Cloisite 20A performed similarly well in improving the flame retardancy of the model compounds, whilst MAX CT 4260 performed slightly worse, although still better than when unmodified Cloisite 116 or no bentonite at all was used.

To contextualize these findings with the actual stability of the resulting composites, the degradation of dumbbell-shaped samples during artificial weathering was monitored by ATR-IR-spectroscopy (the spectra can be found in the Appendix A) using the carbonyl index (CI) as an indicator (Figure 6). Up to a weathering duration of 1500 h, the polymer oxidation on the sample surface showed no difference for the model compounds. From this point on, a clear divergence was evident between the OLS composites on the one hand and the bentonite-free as well as the unmodified bentonite-containing formulation on the other hand. The latter group did not exhibit significant oxidation in the period of observation, whereas the former depicted an exponentially increasing degree of oxidation. The three OLSs substantially differed from each other in terms of their rates of degradation. Cloisite 20A led to the strongest oxidation and Dellite CW9 only showed minor oxidative progress. All the previously analyzed data for Cloisite 20A implicated a fast degradation of the respective composite. The data concerning Dellite CW9, however, suggested a less severe impact on compound stability, most likely due to its smaller pores and smaller specific surface area along with shorter interlayer distances.

Dellite CW9 provides an advantage over the other OLSs concerning its use in flame-retardant cable composites, having the lowest impact on long-term photostability whilst simultaneously imparting strong flame retardancy. It has been proven that the resulting compound stability can be successfully estimated upfront from the pristine bentonite products. Furthermore, it is confirmed that the surface properties of bentonite are of utmost importance to the compound stability, and the corresponding properties are amongst the most important for achieving high long-term photostability.

Additionally, the role of the elimination products originating from the organic modifier [15,33] as a primary influence of compound destabilization for the investigated systems was examined. Thermal stress during melt compounding can lead to the formation of vinylic products through Hofmann elimination, which Dintcheva et al. demonstrated for a nanocomposite of PE and organo-modified montmorillonite for processing temperatures of 180 °C and above. [33] They found an increased concentration of organic compounds via IR spectroscopy between 889 and 922 cm^−1^ for the nanocomposites, corresponding to vinylic elimination products of the organo-modifier. No such elevated concentrations were found by FTIR spectroscopy under the processing conditions applied in this study (up to 165 °C for melt compounding and 190 °C for injection molding), leading to the conclusion that at least under these processing conditions, Hofmann elimination products do not significantly contribute to the decreased long-term photostability of the tested nanocomposites. Additionally, the OLS with the highest overall as well as the highest additional surface-borne modifier content, Dellite CW9, exhibited the lowest degree of oxidation in the weathering experiment, supporting the prior claim.

To enhance the understanding of the influence of the bentonite grade on the corresponding composite long-term photostability, the content of the photocatalytic species of the products was studied using SEM-EDX. From the EDX spectra attached in the Appendix A, the iron content was extracted and normalized to the respective silicon content, resulting in the relative iron content. To estimate the iron content in the actual bentonite products containing the organic modifier, the value was further adjusted by the amount of residue obtained via thermal gravimetry at 800 °C, emulating the inorganic content. These values for the relative iron content in each bentonite are plotted against their respective CI after artificial weathering in Figure 7. The iron content in the native bentonites was obtained by normalization by the Si content and was, with a value between 4.50 and 4.94 at%, comparable for all samples. The iron content in the processed organo-modified bentonites that were utilized for the composite preparation remained the same for unmodified Cloisite 116 but decreased to a comparable 2.8–3.0 at% for the three OLSs, resulting in vastly differing CIs during weathering. Whilst efforts to decrease the destabilizing impact of organically modified silicates utilizing a metal deactivator have been made in the past [20], the positive impact of the additive reported in this work might be due to the low degree of dispersion of the OLS in the pure PE matrix. Contrary to these results, other studies reveal iron to not be primarily responsible for the decrease in long-term stability [34] or observe no significant effect of the iron content on long-term stability. [18] In this study, it was confirmed that the higher amount of iron imparted into the composite by Cloisite 116 alone does not deteriorate the long-term photostability. The same is valid for the OLSs, where no primary correlation between the iron content and the amount of oxidation could be observed, implying that the photocatalytic effect of iron has a lower impact than the surface interactions stated above.

## 4. Conclusions

Different bentonite grades were analyzed by gas adsorption, thermal gravimetry, SEM-EDX, XRD, and TEM to reveal the impact of bentonite properties on the long-term photostability during weathering of bentonite polyolefin nanocomposites. The results indicate that photolytic stability strongly depends on the specific way the organo-modification is applied to the bentonite, which, in turn, determines key properties of the bentonite composite, i.e., its intercalation distance and specific surface area. Based on this approach, the impact of different OLS grades on the weathering stability of bentonite composites can be predicted based on their surface properties.

Pristine Dellite CW9 showed a bimodal interlayer distance distribution, leading to a lowered degree of intercalation and, subsequently, a smaller interface between the filler and polymer. Nitrogen gas adsorption for this bentonite grade revealed a low specific surface area as well as smaller pores, owing to an increased amount of free organo-modifier, decreasing the likelihood of stabilizer adsorption. The corresponding composite displayed the highest long-term photostability amongst all tested OLS nanocomposites, whereas Cloisite 20A, with its high specific surface area and increased pore size, facilitated stabilizer adsorption, resulting in a nanocomposite displaying the lowest photostability. However, the flame-retardant capabilities, as reviewed by cone calorimetry, showed a similarly good performance for all OLS nanocomposites, barring MAX CT 4260 with a higher deviation in the results, owing to the subpar dispersion and subsequent partial agglomeration of the filler on the composite surface.

The experimental findings indicate that the filler–polymer interface with its localized interactions with the stabilizers seems to be the key factor for nano-dispersed OLS composite photostability. However, the common hypothesis of the degradation of the organo-modifier via Hofmann elimination seems to be of minor importance under the processing conditions applied in this study. The impact of photocatalytic iron in bentonite was confirmed not to be the primary source of reduced composite stability. Another common hypothesis for the lowered photostability of bentonite nanocomposites, the photocatalytic polymer degradation imparted by transition-metal contaminants like iron, seemed to not be of importance compared with interactions like stabilizer adsorption on the filler surface.

Due to the findings of the interactions near the filler–polymer interface being essential for polymer stability, an evidence-based estimation for bentonite products in polymer composites with enhanced stability can be established without explicitly testing the resulting compound stability. Furthermore, these findings enable concise efforts in the future for novel designs of the said interface to tailor the interaction that adds the most to the destabilization of OLS composites. Finally, this will provide flame-retardant OLS composites, including formulations to be used in cable manufacturing with increased long-term photostability.

## Figures and Tables

**Figure 1 polymers-16-00535-f001:**
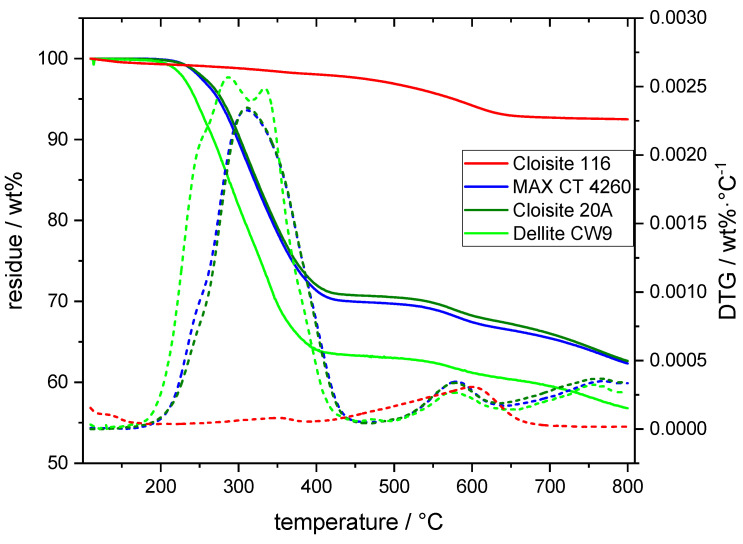
TG and DTG curves for OLS and unmodified bentonite samples. TG and DTG curves are denoted by the solid and dashed lines, respectively.

**Figure 2 polymers-16-00535-f002:**
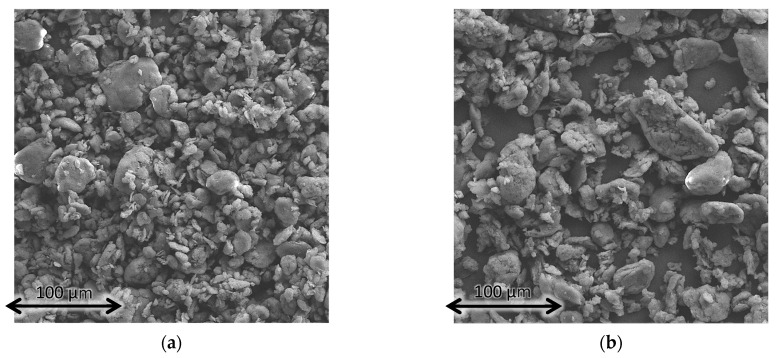
SEM micrographs at 500× magnification, covering an area of 270 × 270 µm^2^ for (**a**) OLS Dellite CW9, (**b**) OLS MAX CT 4260, (**c**) OLS Cloisite 20A, and (**d**) unmodified bentonite Cloisite 116.

**Figure 3 polymers-16-00535-f003:**
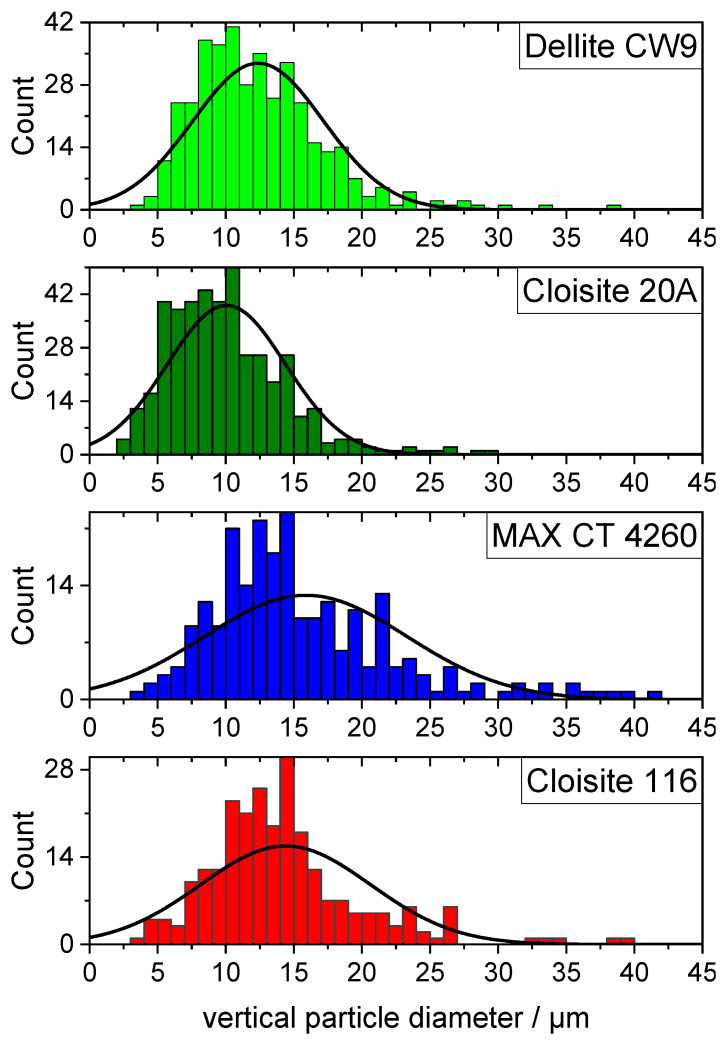
Particle size distribution of bentonite sample inferred by SEM imaging. The solid lines represent distribution curves and were fitted to the data evaluated by image analysis.

**Figure 4 polymers-16-00535-f004:**
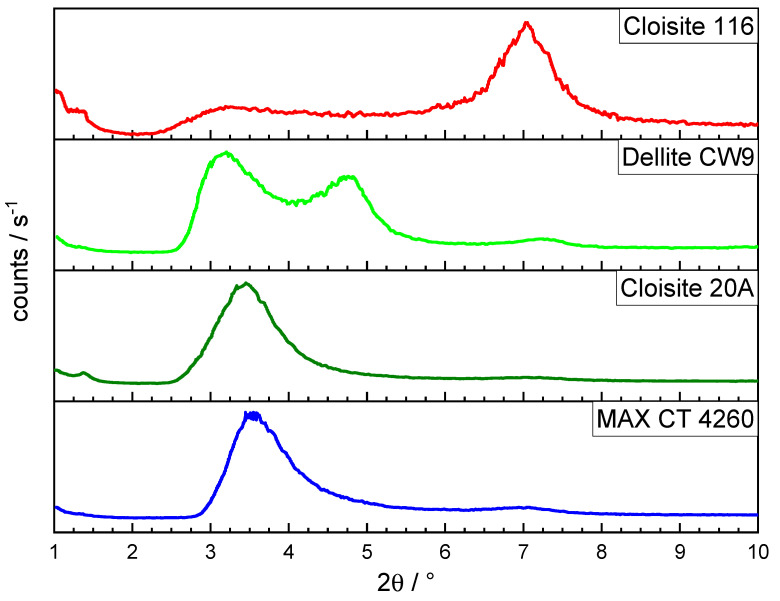
Results of XRD analysis depicting *d_001_* distances of bentonite products.

**Figure 5 polymers-16-00535-f005:**
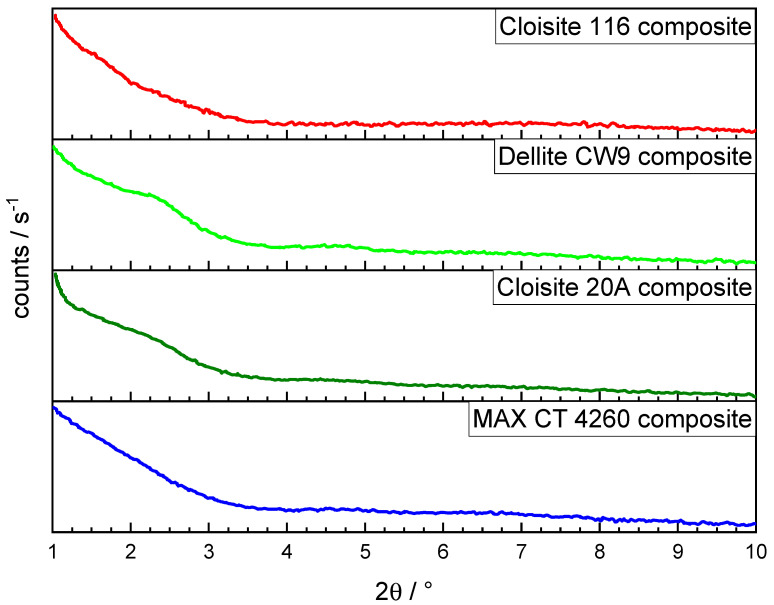
XRD analysis of bentonite-filled nanocomposites.

**Figure 6 polymers-16-00535-f006:**
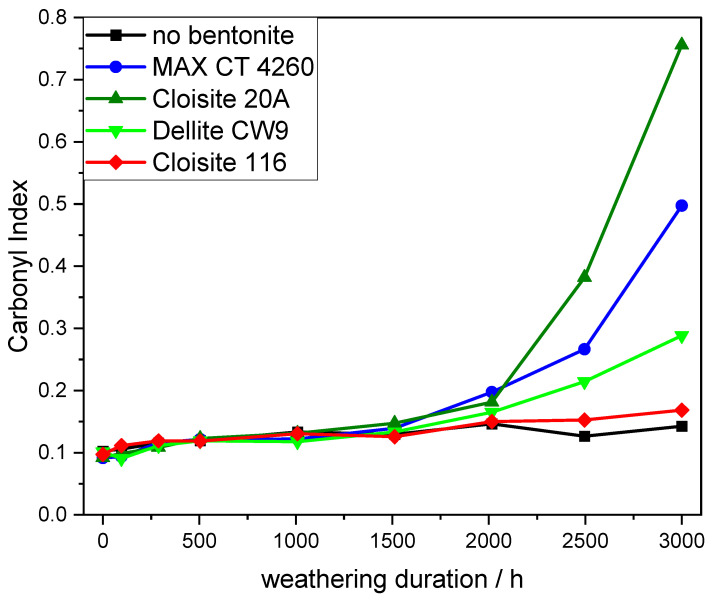
Carbonyl index during weathering of model composites.

**Figure 7 polymers-16-00535-f007:**
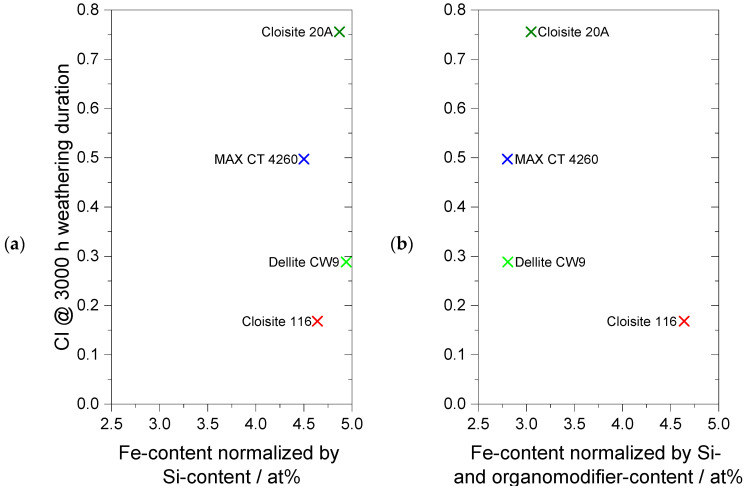
Iron content obtained by SEM-EDX analysis, normalized by (**a**) Si content and (**b**) Si content and organo-modifier content obtained by TGA, each with corresponding CI and the end of weathering.

**Table 1 polymers-16-00535-t001:** N_2_ gas adsorption results of OLS and unmodified bentonite samples.

Bentonite	Specific Surface Area /m^2^·g^−1^	Median Pore Diameter /nm	Cumulative Pore Volume /cm^3^·g^−1^
Cloisite 116	29.79	5.992	0.0753
MAX CT 4260	15.10	14.67	0.0593
Cloisite 20A	18.51	15.46	0.0628
Dellite CW9	15.40	11.65	0.0363

**Table 2 polymers-16-00535-t002:** Characteristic values of particle sizes of bentonite products.

Bentonite	Maximum of Distribution/µm	FWHM/µm
Cloisite 116	14.45	14.78
MAX CT 4260	15.88	17.43
Cloisite 20A	10.02	10.17
Dellite CW9	12.38	11.09

**Table 3 polymers-16-00535-t003:** Peak heat release rate (PHRR), total heat and smoke release (THR/TSR), maximum average rate of heat emission (MARHE), and residue of model composites obtained by cone calorimetry.

Bentonite	PHRR/kW·m^−2^	THR/MJ·m^−2^	TSR/m^2^·m^−2^	MARHE/kW·m^−2^	Residue/%
No bentonite	182 ± 22	56.4 ± 5.3	279 ± 115	95.5 ± 14.7	42.6 ± 1.3
MAX CT 4260	102 ± 15	50.6 ± 9.2	203 ± 85	59.4 ± 12.2	43.9 ± 3.4
Cloisite 20A	100 ± 5	47.3 ± 1.8	131 ± 17	57.5 ± 2.4	44.9 ± 0.1
Dellite CW9	92 ± 7	43.6 ± 4.6	135 ± 53	52.9 ± 6.1	46.0 ± 0.8
Cloisite 116	113 ± 1	52.9 ± 1.1	270 ± 56	74.3 ± 1.4	44.4 ± 1.0

## Data Availability

The data presented in this study are available upon request from the corresponding author.

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
