# Peer review of "Strategies to Cope with Inferior Long-Term Photostability of Bentonite Polyolefin Nanocomposites"

_polymers, 2024, doi:10.3390/polym16040535_

Round 1
Reviewer 1 Report
Comments and Suggestions for Authors
Comments
In this research, the author provided some strategies to improve the long-term photostability of the Bentonite polyolefin nanocomposites. In my view, the following issues need to be addressed before consideration for publication:
1. The author mentioned the dehydroxylation of the montmorillonite, but the author didn’t explain the principle of its.
2. The author didn’t explain why Cloisite 20A led to the strongest oxidation and Dellite CW9 only showed minor oxidative progress.
3. The author proposed iron content didn’t have significant effect on long-term photostability.Please briefly describe the reasons for it.
4. The author presented the barrier properties were related to sufficient dispersion of the bentonite platelets.But he did not explain the mechanism of dispersion.
Comments on the Quality of English Language
Minor editing of English language required
Author Response
Thank you for your valuable feedback. The comment on iron content was especially helpful to improve the overall quality of this manuscript! For each of your comments, we have adapted the manuscript in the following way:
1. The author mentioned the dehydroxylation of the montmorillonite, but the author didn’t explain the principle of it: Clarified structural Si-OH groups are being dehydroxylated
- The author didn’t explain why Cloisite 20A led to the strongest oxidation and Dellite CW9 only showed minor oxidative progress: Reiterated the importance of the surface properties of organomodifed bentonite products due to their interactions with stabilizers and thus their differing impact on destabilization
- The author proposed iron content didn’t have significant effect on long-term photostability. Please briefly describe the reasons for it: Rephrased so that it becomes clearer that there is a vast difference in photostability while the iron content for OLS is very similar. Additionally, emphasized that the highest amount of iron in composites (for unmodified Cloisite 116) shows the lowest degree of degradation.
- The author presented the barrier properties were related to sufficient dispersion of the bentonite platelets. But he did not explain the mechanism of dispersion: Added explanation for overcoming the entropy barrier for dispersion employing compatibilizers on the polymer side as well as amphiphilic organic modifier in the interlayers of OLS
Reviewer 2 Report
Comments and Suggestions for Authors
The manuscript “Strategies for improved long-term photostability of bentonite polyolefin nanocomposites” is an interesting work, but before its acceptance, it needs some improvements.
The title induces us to think that by adding the additives, such as MAX CT 4260, Cloisite 20A, Dellite CW9, Cloisite 116, the photostability of OLS is increased. Based on data reported in Figure 6, neat OLS (sample named: no bentonite) shows the lowest Carbonyl Index, i.e. oxygen-containing groups accumulations, suggesting the highest photostability. The title of the paper must be changed taking into account the corrected interpretation of reported data.
Further, Figure 6 shows the calculated carbonyl index and there is not reported the original FTIR spectra. To understand better the occurrence of the phenomena, it is requested to show the FTIR spectra, also as supplementary information.
To confirm the hypothesis regarding the improved oxidative resistance of OLS-based composites it will be useful to add TGA analysis.
It seems that the authors attributed the accelerated degradation (higher CI) of composites to the presence of iron ions in the structure of bentonite rather than to the Hoffman degradation of the organo-modifier, see comments of Figure 7, but the differences between Cloisete 20A (organo-modified bentonite) and Closite 116 (some bentonite, without organo-modifier) is only the presence of organo-modifier. To support the author's hypothesis, there will need to be elemental analysis, for example, EDX, of the native bentonite structure. If this analysis has been done, please show the results; maybe this analysis is done considering the x-axis values in Figure 7.
The photo-oxidation resistance of clay-containing composites was widely documented in the scientific literature and here reported references do not reflect the most important papers on this topic. There is cited a limited number of papers regarding this very important issue, especially, from of industrial point of view.
Author Response
Thank you for your valuable feedback. The comment on iron content was especially helpful to improve the overall quality of this manuscript! For each of your comments, we have adapted the manuscript in the following way:
I. The title (Strategies for improved long-term photostability of bentonite polyolefin nanocomposites) induces us to think that by adding the additives, such as MAX CT 4260, Cloisite 20A, Dellite CW9, Cloisite 116, the photostability of OLS is increased. Based on data reported in Figure 6, neat OLS (sample named: no bentonite) shows the lowest Carbonyl Index, i.e. oxygen-containing groups accumulations, suggesting the highest photostability. The title of the paper must be changed taking into account the corrected interpretation of reported data. - Adjusted title to better represent the essence of the data (Strategies to cope with inferior long-term photostability of bentonite polyolefin nanocomposites)
II. Figure 6 shows the calculated carbonyl index and there is not reported the original FTIR spectra. To understand better the occurrence of the phenomena, it is requested to show the FTIR spectra, also as supplementary information. - Added all spectral data to supplementary data
III. To confirm the hypothesis regarding the improved oxidative resistance of OLS-based composites it will be useful to add TGA analysis.
- TGA results might not be expedient. The Subject of the paper is increased photostability, which is not necessarily correlated to the oxidative process during TGA, especially due to increased temperatures allowing for changed kinetics and entirely new degradation reactions compared to degradation at conditions that are within expectations for cables to experience (outside of fires and the like)
- However, residues from cone calorimetry were added to the respective table and briefly discussed since they can express similar results as TGA under oxygen.
IV. It seems that the authors attributed the accelerated degradation (higher CI) of composites to the presence of iron ions in the structure of bentonite rather than to the Hoffman degradation of the organo-modifier, see comments of Figure 7, but the differences between Cloisite 20A (organo-modified bentonite) and Cloisite 116 (some bentonite, without organo-modifier) is only the presence of organo-modifier. To support the author's hypothesis, there will need to be elemental analysis, for example, EDX, of the native bentonite structure. If this analysis has been done, please show the results; maybe this analysis is done considering the x-axis values in Figure 7.
- Rephrased, to emphasize that there was no correlation found between iron content and degree of degradation (highest amount in Cloisite 116 remains the most stable composite, while OLS show vastly different degradation behavior with comparable iron contents)
- Rephrased so that it is more obvious to the reader that elemental analysis was done by SEM-EDX and the resulting iron content is normalized using various means. Additionally, another graph is employed to show the difference between two normalization steps. Furthermore, EDX spectra are added to the supplementary data.
V. The photo-oxidation resistance of clay-containing composites was widely documented in the scientific literature and here reported references do not reflect the most important papers on this topic. There is cited a limited number of papers regarding this very important issue, especially, from of industrial point of view.
- Added 8 references to better reflect the subject
Round 2
Reviewer 2 Report
Comments and Suggestions for Authors
The paper can be accepted in this form